# Transplacental Transfer of SARS-CoV-2 Receptor-Binding Domain IgG Antibodies from Mothers to Neonates in a Cohort of Pakistani Unvaccinated Mothers

**DOI:** 10.3390/biomedicines11061651

**Published:** 2023-06-06

**Authors:** Steve Harakeh, Ihsan Alam Khan, Gulab Fatima Rani, Muhammad Ibrahim, Aysha Sarwar Khan, Mohammed Almuhayawi, Rajaa Al-Raddadi, Addisu D. Teklemariam, Mohannad S. Hazzazi, Waleed M. Bawazir, Hanouf A. Niyazi, Turki Alamri, Hatoon A. Niyazi, Yasar Mehmood Yousafzai

**Affiliations:** 1King Fahd Medical Research Center, King Abdulaziz University, Jeddah 21589, Saudi Arabia; 2Yousef Abdul Latif Jameel Scientific Chair of Prophetic Medicine Application, Faculty of Medicine, King Abdulaziz University, Jeddah 22230, Saudi Arabia; 3Department of Hematology, Institute of Pathology and Diagnostic Medicine, Khyber Medical University, Peshawar 25100, Pakistan; 4Department of Pathology, Swat Medical College, Swat 19200, Pakistan; 5Department of Clinical Microbiology and Immunology, Faculty of Medicine, King Abdulaziz University, Jeddah 21589, Saudi Arabiahniyazi@kau.edu.sa (H.A.N.); 6Community Medicine Department, Faculty of Medicine, King Abdulaziz University, Jeddah 21589, Saudi Arabia; 7Department of Biological Sciences, Faculty of Science, King Abdulaziz University, Jeddah 21589, Saudi Arabia; 8Department of Medical Laboratory Sciences, Faculty of Applied Medical Sciences, King Abdulaziz University, Jeddah 22254, Saudi Arabiawbawazir@kau.edu.sa (W.M.B.); 9Hematology Research Unit, King Fahd Medical Research Center, King Abdulaziz University, Jeddah 22254, Saudi Arabia; 10Family and Community Medicine Department, Faculty of Medicine in Rabigh, King Abdulaziz University, Jeddah 21589, Saudi Arabia; 11Rehman Medical Institute, Hayatabad Phase-V, Peshawar 25600, Pakistan

**Keywords:** transplacental passage, SARS-CoV-2, RBD IgG, transmission from mothers to neonates, Pakistani

## Abstract

The presence of COVID-19 antibodies in the maternal circulation is assumed to be protective for newborns against SARS-CoV-2 infection. We investigated whether maternal COVID-19 antibodies crossed the transplacental barrier and whether there was any difference in the hematological parameters of neonates born to mothers who recovered from COVID-19 during pregnancy. The cross-sectional study was conducted at the Saidu Group of Teaching Hospitals, located in Swat, Khyber Pakhtunkhwa. After obtaining written informed consent, 115 healthy, unvaccinated mother-neonate dyads were included. A clinical history of COVID-19-like illness, laboratory-confirmed diagnosis, and contact history were obtained. Serum samples from mothers and neonates were tested for SARS-CoV-2 anti-receptor-binding domain (anti-RBD) IgG antibodies. Hematological parameters were assessed with complete blood counts (CBC) and peripheral blood smear examinations. The study population consisted of 115 mothers, with a mean age of 29.44 ± 5.75 years, and most women (68/115 (59.1%)) were between 26 and 35 years of age. Of these mothers, 88/115 (76.5 percent) tested positive for SARS-CoV-2 anti-RBD IgG antibodies, as did 83/115 (72.2 percent) neonatal cord blood samples. The mean levels of SARS-CoV-2 IgG antibodies in maternal and neonatal blood were 19.86 ± 13.82 (IU/mL) and 16.16 ± 12.90 (IU/mL), respectively, indicating that maternal antibodies efficiently crossed the transplacental barrier with an antibody transfer ratio of 0.83. The study found no significant difference in complete blood count (CBC) parameters between seropositive and seronegative mothers, nor between neonates born to seropositive and seronegative mothers.

## 1. Introduction

According to the published literature, up to 25% of COVID-19 cases are reported in women of reproductive age [1]. Pregnant women are at a higher risk of morbidity and mortality due to systemic infections compared to non-pregnant women of the same age [2]. Due to its systemic nature and associated severity, COVID-19 infection has been shown to impact fetal and neonatal development with undesirable outcomes such as the increased risk of abortion, preterm delivery, low birth weight, and stillbirth [3,4]. The mechanism of such complications can be due to various reasons. Studies have shown that COVID-19-induced inflammation and vasculitis in the placenta can lead to impaired blood flow and reduced oxygen delivery to the fetus. In addition to placental damage, COVID-19 can also cause a systemic inflammatory response in the mother’s body, which may affect fetal development and contribute to fetal distress. Thromboembolism, one of the most recognized complications of COVID-19, can increase the risk of placental abruption, and cause life-threatening bleeding for both the mother and the fetus. It is not known whether maternal COVID-19 can cause long-term complications in the fetus, particularly in the hematopoietic system.

Immunoglobulin G (IgG) is the only class of antibodies that has the capacity to be transferred from mother to fetus through the placenta, facilitated by the Fc receptor (FcRn) expressed on the syncytiotrophoblast cells of the placenta [5]. These passively derived antibodies from mothers play a key role in neonatal immunity against common infections. Maternal antibody responses to COVID-19 infection during pregnancy and transplacental antibody transfer may be important for the management and vaccination of neonates and pregnant women, respectively [6]. Maternal IgG antibodies developed against severe acute respiratory syndrome coronavirus 2 (SARS-CoV-2) infection during pregnancy cross the placental barrier and are protective against SARS-CoV-2 infection in newborns [6]. Transmission ratios of IgG antibodies from COVID-19-infected pregnant women to neonates have been reported to vary between <0.3 and ≥1.55, and there is a significant association between maternal antibodies and umbilical cord antibody concentration [6,7]. Furthermore, a number of reports have shown the absence of antibodies in neonates despite high levels of maternal antibodies [6,7,8]. Given the variation in the perinatal antibody transfer ratio, it has been recommended that the transplacental transfer of antibodies be investigated in different populations [8].

Although fetal distress and death are established complications of maternal COVID-19, little evidence exists on the longer-term effects of maternal COVID-19 on the fetus. The hematological consequences of COVID-19 include thrombocytopenia and thromboembolism. COVID-19 has also been shown to affect hematopoiesis. The infection-induced cytokines result in a state of ‘stress-induced hematopoiesis’ in the hematopoietic stem cells. The viral infection itself can affect hematopoiesis [9].

We asked whether SARS-CoV-2 IgG antibodies cross the placental barrier and can be found in neonatal blood. In addition, we asked whether COVID-19 infection at any given time in pregnancy had negative effects on the hematopoietic systems of mothers and neonates at the time of delivery. In this study, we enrolled mothers at the time of delivery and assessed mothers’ and neonates’ hematological profiles, as well as their SARS-CoV-2 IgG antibodies.

## 2. Materials and Methods

### 2.1. Study Participants

The study was conducted with approval from the Khyber Medical University’s Peshawar ethical committee (letter no.: KMU/IBMS/2021/4860). It was a descriptive cross-sectional study that took place at the Saidu Group of Teaching Hospitals in Swat and the Institute of Pathology and Diagnostic Medicine (IPDM) at Khyber Medical University (KMU) between 28 January and 28 June 2022. Non-probability sampling was used to recruit study participants, which included mothers who were exposed to COVID-19 infection during their first, second, and third trimesters, as well as their neonates on day one. Mothers with a history of hematological disorders, as well as those with diabetes, hypertension, cardiovascular diseases, and other significant comorbidities, vaccinated mothers, previously exposed mothers before pregnancy, and non-consenting participants, were excluded from the study.

The objectives of the study were clearly explained to participants before seeking written informed consent and taking a complete clinical history. The clinical history questionnaire gathered information on COVID-19 PCR test results, a history of contact with a positive family member, and symptoms of COVID-19 (flu-like illness) during pregnancy.

### 2.2. Sample Collection and Processing

At the time of delivery, a 3 mL blood sample was collected from each mother using an EDTA tube (Lot#210801, Golden Vac, Shenyang City, Liaoning Province, China) and a gel tube/Clot Activator (Lot#210801, Golden Vac, Shenyang City, Liaoning Province, China), as well as from their neonates via the umbilical cord. The blood sample collected in the gel tube was centrifuged at 4000 rpm for 5 min to extract serum, which was subsequently stored at −80 °C until further analysis. The sera were thawed according to the standard protocol before quantifying SARS-CoV-2 IgG antibodies using an enzyme-linked immunosorbent assay (ELISA). The whole blood samples of mothers and neonates collected in EDTA tubes were analyzed for complete blood counts using a 5 part hematology analyzer called ‘Sysmex XN 550’ (Sysmex Corporation, Chuo-ku 651-0073 Kobe, Japan).

### 2.3. Indirect Enzyme-Linked Immunoassay

A quantitative anti-SARS-CoV-2 IgG antibody kit from Vircell (Ref. No. G1032, Lot. No. 20ECOG114, Vircell, Granada, Spain) was used to perform an ELISA following the manufacturer’s guidelines. The indirect ELISA method was used, and the kit had a sensitivity and specificity of 99%. Briefly, all the reagents were brought to room temperature. The plate was removed from the package. A 1/20 dilution of serum samples was prepared by adding 95 μL of sample dilution solution and a patient sample of 5 μL. Except for the control wells, 80 μL of sample dilution solution was added to the test well. Then 20 μL of 1/20 dilution of a serum sample that was prepared earlier was added into the test well that already contained 80 μL of sample dilution solution, making it 100 μL of the solution. Additionally, 100 μL of positive control, 100 μL of duplicate cutoff, and 100 μL of negative control were added to their corresponding wells. These were incubated at 37 °C while covered with a sealing sheet. After incubation, the sealing sheet was removed, and the wells were washed five times with a 0.3 mL washing solution. Subsequently, 100 μL of IgG conjugates were added to all wells and incubated at 37 °C for 30 min while covered with a sealing sheet. After incubation, all the wells were again washed with 0.3 mL washing solution five times. Then, 100 μL of substrate solution was added immediately and incubated for 20 min while protected from light. Finally, 50 μL of stopping solution was immediately added to all the wells, and the result was obtained with a spectrophotometer Biotek Elx machine (Winooski, VT 05404, USA) at 450/620 nm within one hour of the stop solution.

In this study, a cut-off value of <4 was used to classify results as negative for SARS-CoV-2 anti-RBD IgG antibodies, while a value of >6 was considered positive. For values falling between 4 and 6, the results were considered equivocal and repeated. All standards were included in the kit, and samples were run in duplicate to ensure accuracy. To minimize the possibility of detecting non-specific antibodies, negative controls were run using sera collected prior to the COVID-19 pandemic (samples collected in 2018 for unrelated work and stored in the KMU biobank). IgG levels were calculated using MS Excel 2013 (Microsoft, Redmond, WA, USA) based on the absorbance values and the manufacturer’s guidelines. Mothers and neonates with detectable IgG antibodies were considered seropositive, while those without detectable antibodies were deemed seronegative.

### 2.4. Statistical Analysis

The data obtained from the study were analyzed using Statistical Package for Social Sciences (SPSS) software, version 26.0^®^ (IBM Corp., Armonk, NY, USA), a widely used software for statistical analysis. The mean and standard deviation were calculated for continuous numerical variables, while frequency and percentages were calculated for discrete categorical variables. To calculate the SARS-CoV-2 IgG transfer ratio, the cord blood IgG levels were divided by the corresponding maternal blood IgG levels and multiplied by 100. Pearson correlation analysis was used to determine the relationship between maternal and neonatal SARS-CoV-2 IgG antibody levels. The independent sample t-test was used to compare hematological parameters between seropositive and seronegative mothers and between neonates born to seropositive and seronegative mothers. A *p*-value of less than 0.05 was considered statistically significant.

## 3. Demographic and Clinical Characteristics of the Study Participants

In this study, a total of 115 perinatal mother-neonate dyads were included. The mean age of the mothers was 29.44 ± 5.75 years, with the majority of them (*n* = 68, 59.1%) being between 26 and 35 years old. There were 36 (31.3%) mothers in the age range of 15–25 years and 11 (9.6%) mothers above 35 years of age. Most of the women (91.3%) delivered babies at 37–42 weeks of gestation, while only 10 (8.7%) had their delivery before 37 weeks. Among the pregnant women, 70 (60.9%) had a normal vaginal delivery, and 45 (39.1%) had a cesarean section. The mean birthweight of all newborns was 2.01 ± 0.264 Kg, with 68 (59.1%) female babies and 47 (40.9%) male babies born. The majority of the babies (107, 93%) were born with a normal birth weight (2.5–3.5 Kg), while only three (2.6%) had a low birth weight (<2.5 Kg). Most of the newborns (80%) had a normal APGAR score (>7) at 5 min.

Since this study aimed to assess the correlation between maternal IgG antibodies and neonatal IgG levels and their effect on the hematopoietic system of neonates, we enrolled mothers who had been exposed to COVID-19 infection during pregnancy. In this study, four (3.47%) mothers had a positive COVID-19 PCR test during pregnancy, with three (75%) testing positive in the first trimester and one (25%) in the second trimester. Some of these mothers (*n* = 35, 30.43%) had a positive contact history with a diagnosed COVID-19 patient, with seventeen (48.57%) having a positive contact history in their first trimester, fifteen (42.85%) in the second trimester, and only three (8.57%) in the third trimester. The remaining 76 (66.08%) mothers had symptoms consistent with a COVID-19 infection during pregnancy. (Table 1).

### 3.1. Anti-SARS-CoV-2 IgG Levels in Mothers and Neonates

In our study, out of the 115 mother-neonate dyads, 88 (76.5%) mothers and 83 (72.2%) neonates tested positive for SARS-CoV-2 IgG antibodies (Figure 1). The mean SARS-CoV-2 IgG antibody level in mothers was 19.86 ± 13.82 (IU/mL), while the mean level in neonates was 16.16 ± 12.90 (IU/mL). The mean difference between maternal IgG and neonatal IgG was 3.75 ± 6.104 (*p*-value < 0.001, paired samples *t*-test) (Figure 1b). The mean IgG transfer ratio, calculated as the mean maternal IgG divided by the mean neonatal IgG, was 0.813 (±0.214).

A strong positive correlation (*r* = 0.806) was observed between maternal and neonatal antibodies based on bivariate Pearson correlation analysis, which was statistically significant with a *p*-value of <0.001 (Figure 1c).

### 3.2. Comparison of Hematological Parameters between Seropositive and Seronegative Neonates

Assuming that maternal SARS-CoV-2 infection during pregnancy could potentially affect the hematopoiesis of the fetus, we conducted a comparison between the hematological parameters of neonates born to seropositive mothers and those born to seronegative mothers. Through independent sample t-tests, we found no statistically significant difference in any of the hematological parameters, including red cells, white cells, and platelets. (Table 2).

### 3.3. Comparison of Hematological Parameters of Seropositive and Seronegative Mothers

The acute inflammation phase of SARS-CoV-2 infection has been shown to have a significant impact on the hematopoietic system. However, it is not clear if the infection during pregnancy causes any sustained effect on maternal hematological parameters during the post-acute phase. Therefore, this study aimed to investigate if there is any long-term impact of SARS-CoV-2 infection during pregnancy on maternal hematological parameters. Hematological parameters were compared between seropositive mothers and seronegative mothers using an independent sample *t*-test. The analysis revealed no statistically significant difference in any of the hematological parameters, including red cells, white cells, and platelets. (Table 3) Top of Form.

## 4. Discussion

In this study, a total of 115 pairs of perinatal mothers and neonates were enrolled, and their SARS-CoV-2 antibodies and hematological parameters were quantified. This study, conducted on a Pakistani cohort, provides novel findings regarding the transfer of SARS-CoV-2 IgG antibodies across the placenta. It was found that there is an efficient transfer of these antibodies from mothers to neonates. Furthermore, the study demonstrates that there is no significant difference in hematological parameters between neonates born to SARS-CoV-2 antibody-seropositive mothers and those born to seronegative mothers. Additionally, the findings suggest that there is no significant difference in the hematological parameters of neonates who received maternally transferred SARS-CoV-2 antibodies compared to those who did not receive them. These results contribute to our understanding of the impact of maternal SARS-CoV-2 infection on neonatal outcomes and hematological parameters.

COVID-19 infections were shown to infect everyone worldwide regardless of age, including women of reproductive age [1]. Several physiological and immunological changes occur during pregnancy, which predispose pregnant women to viral infections, including COVID-19 [10]. In our study, the majority of the neonates were born via normal vaginal delivery, but the number of pregnant women undergoing cesarean sections was still high (39.1%). These results are consistent with the findings reported by Yuan et al. [11]. The most plausible explanation is that most patients were recruited in hospital settings where more complicated pregnancies were present for delivery. The rate of cesarean sections varies in different countries, with a higher rate in developed countries [12]. Nevertheless, the higher proportion of cesarean sections in our study population is outside the scope of this study.

In our study, more than 3/4th of unvaccinated mothers were positive for SARS-CoV-2 IgG antibodies. This high seroprevalence in the unvaccinated population is not surprising. Large population-based surveys, both from Pakistan and globally, have shown that the majority of unvaccinated populations have experienced SARS-CoV-2 infection at some stage [13,14]. In rural areas of Pakistan, where COVID-19 lockdown measures were less stringent, widespread testing was not readily available, and social interactions were uninhibited, it is logical to expect a high prevalence of SARS-CoV-2 infection. In our recent unpublished work, we observed a seroprevalence of 93% among the unvaccinated elderly Pakistani population.

We based our analysis on the assumption that these mothers contracted the SARS-CoV-2 infection during pregnancy. There is ample evidence to suggest the presence of SARS-CoV-2 antibodies in the circulation for several months post-infection, followed by a gradual decline [15,16]. Therefore, the epidemiological evidence of the high incidence of COVID-19 infection in our study area, positive contact history, COVID-19-like symptoms, and seropositivity at the time of parturition were considered surrogate markers of SARS-CoV-2 infection during pregnancy [17,18].

The majority of the neonates born to seropositive mothers also had SARS-CoV-2 IgG antibodies in their cord blood. The presence of maternal SARS-CoV-2 IgG in neonates has been shown to provide immunity to infants for up to 6 months of age [19], and is critical for a newborn’s immunity against common infections [20]. However, in our study, the absence of antibodies was seen in 5/88 (5.6%) neonates with seropositive mothers. A number of previous studies have observed similar results, ranging from 10.9% (8/73) [7], 13% (11/83) [6], and 25.4% (50/67) [21].

In our study, the IgG transfer ratio of SARS-CoV-2 antibodies from mother to neonates was 0.813 in seropositive dyads. A number of recent studies have reported varying degrees of antibody transfer ratios, ranging from <1 to >1. [7,22,23,24]. The transmission of SARS-CoV-2 antibodies during the third trimester of pregnancy is significantly more efficient than the 3rd trimester compared with the transfer of antibodies specific to pertussis- and influenza-specific antibodies [25]. Moreover, the levels of SARS-CoV-2 neutralizing IgG antibodies developed at the initial stages of pregnancy remain elevated at later stages, with a high tendency for transplacental transmission to neonates [26]. The exact reason for this absence of transplacental transmission is not clearly understood. However, this could be due to technical errors in performing the ELISA. Other, more plausible mechanisms have been postulated. In one study, fetal IgG concentrations correlated positively with higher maternal antibody levels and a longer duration between maternal infection and time to delivery [27]. In our study, we found a strong correlation between maternal and fetal antibody levels. However, there were paradoxical cases of high levels of maternal antibodies in seronegative neonates. Lower antibody transfer ratios were seen in mothers with third trimester infection and linked with altered IgG glycosylation [28] and placental pathology [5,29]. Some of the reported placental pathologies include placentitis, malperfusion, thrombosis, and fibrin deposition and are associated with adverse outcomes [29,30]. We did not assess the placental pathology or the IgG subclasses in our study.

IgG antibodies are capable of transplacental passage, as shown, but the possibility of neonatal infection cannot be ruled out based solely on neonatal IgG. It is plausible that at the time of maternal COVID-19 infection, the developing fetus might also get exposed to the virus through vertical transmission. Although a rare possibility, the reported findings of some studies point towards the possible transplacental transmission of COVID-19 [31,32].

COVID-19 infection is shown to affect the hematological parameters, and if it occurs in pregnancy, there is a risk of impacting the developing hematopoietic system of the fetus. Clinical studies from China and Vietnam have reported lymphocytosis, lymphopenia, and abnormal platelet counts in neonates exposed to COVID-19 infection [33,34]. The effect of maternal COVID-19 infection on fetal hematopoiesis is largely unknown. In this study, we did not notice any difference in the hematological parameters of neonates with SARS-CoV-2 antibodies or born to seropositive mothers. We also did not observe any difference between the hematological parameters of seropositive and seronegative mothers. Similar findings were reported by Murphy et al. in their study on neonates born to mothers following COVID-19 infection in pregnancy [4].

In this study conducted on a Pakistani cohort of mother-neonate dyads, we observed efficient transfer of SARS-CoV-2 antibodies across the placental barrier. However, in a minority of cases, we did not observe antibody transfer in fetal blood, even in cases with high maternal antibody levels. Despite this, we found no significant difference in the fetal blood counts of neonates born to mothers who recovered from the SARS-CoV-2 infection during pregnancy. Additionally, there was no significant difference in the blood counts between seropositive and seronegative mothers, indicating recovery of the hematopoietic system.

It is important to acknowledge the limitations of this clinical study. Firstly, most participants did not have laboratory confirmation of SARS-CoV-2 infection, which may introduce uncertainty in the interpretation of the results. Secondly, a detailed assessment of the reasons for the absence of antibody transfer was not conducted, and further investigation is warranted in this regard. The observation of the lack of antibody transfer in a significant minority of cases has potential policy implications, as current recommendations for neonates rely on passive immunity from mothers without direct vaccination. Thirdly, IgG subclass testing and laboratory experiments on the infection protection conferred by placentally transferred antibodies were not conducted. However, it is assumed that the repertoire of SARS-CoV-2 IgG includes anti-RBD neutralizing antibodies.

Further research is needed to address these limitations and gain a deeper understanding of the transfer and protective capacity of SARS-CoV-2 antibodies across the placenta. 

## 5. Conclusions

The findings of the study indicate that the transmission of anti-SARS-CoV-2 IgG antibodies from mothers to neonates through the placenta is efficient. Additionally, the study reveals that neonates born to mothers who experienced uncomplicated COVID-19 infection during pregnancy did not show significant differences in hematological parameters compared to neonates born to uninfected mothers. This suggests that maternal infection did not have a sustained impact on these parameters in the post-acute phase. Furthermore, no significant differences were observed in the hematological parameters between seropositive and seronegative mothers.

## Figures and Tables

**Figure 1 biomedicines-11-01651-f001:**
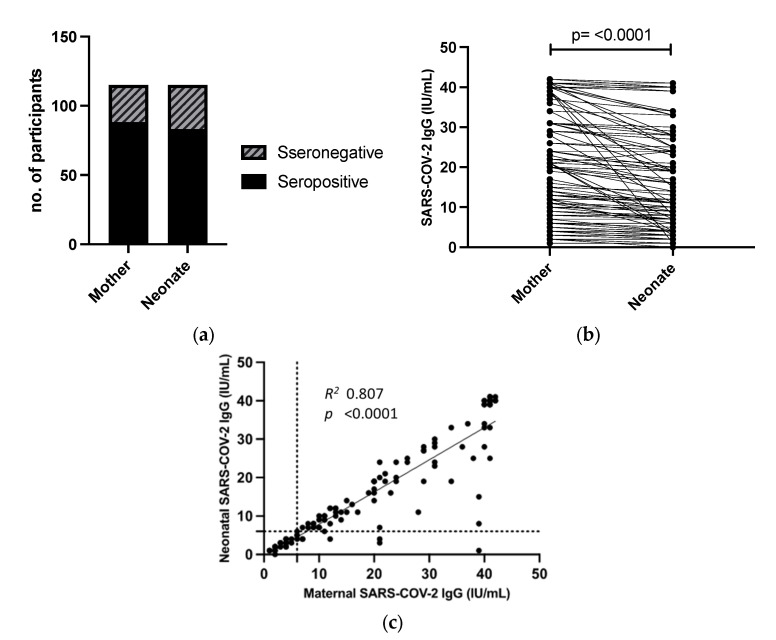
(**a**) SARS-CoV-2 IgG-level seropositivity in mothers and neonates. (**a**) A bar chart showing the number of seropositive mothers and neonates. (**b**) Paired-samples analysis of levels of SARS-CoV-2 antibody levels in mother-neonate dyads, analyzed with a paired-samples *t*-test. (**c**) Scatter plot of maternal and neonate SARS-CoV-2 RBD IgG antibodies (*n* = 115). SARS-CoV-2 RBD IgG antibody levels were measured in the mother’s and neonate’s samples. Levels from each mother-neonate pair were plotted on a scatterplot using Pearson correlation analysis. *X*-axis = maternal SARS-CoV-2 RBD IgG antibodies, IU/mL; *Y*-axis = neonatal SARS-CoV-2 RBD IgG antibody levels, IU/mL; dashed line = cut-off for positivity; and R^2^ = coefficient of correlation.

**Table 1 biomedicines-11-01651-t001:** Demographic and clinical characteristics of the study participants (*n* = 115).

Characteristics	Mean	±SD
Mother’s age (years)	29.44	±5.75
Baby weight (kilograms)	2.01	±0.26
APGAR score	7.50	±1.36
Characteristics	Frequency	Percentage
Mother age group		
15–25 years	36	31.3%
26–35 years	68	59.1%
>35 years	11	9.6%
Gestational age		
<36 weeks	10	8.7%
37–42 weeks	105	91.3%
Mode of delivery		
Normal vaginal delivery	70	60.9%
Cesarean section	45	39.1%
Baby Gender		
Female	68	59.1%
Male	47	40.9%
Baby weight		
<2.5 kg	3	2.6%
2.5–3.5 kg	107	93%
>3.5 kg	5	4.3%
APGAR Score		
0–3	1	0.9%
4–6	22	19.1%
>7	92	80%
Maternal COVID-19 history		
PCR +ve	4	3.47%
Contact history with PCR +ve patient	35	30.43%
COVID-19 symptoms (flu-like illness) during pregnancy	76	66.08%

**Table 2 biomedicines-11-01651-t002:** Hematological parameters of seropositive and seronegative neonates.

Hematological Parameters	Neonatal SARS-CoV-2 IgG Antibodies (*n* = 115)		*p*-Value *
	Seropositive (*n* = 888) (Mean ± SD)	Seronegative (*n* = 2727) (Mean ± SD)	
Hemoglobin (g/dL)	15.43 ± 2.09	15.81 ± 1.63	0.360
MCH (pg)	35.39 ± 2.13	35.69 ± 2.19	0.510
MCHC (g/dL)	34.85 ± 1.71	35.053 ± 2.01	0.597
MCV (fL)	101.73 ± 5.35	102.0 ± 5.10	0.801
Neutrophils (×10^3^/µL)	5.94 ± 2.36	6.19 ± 1.86	0.592
Lymphocytes (×10^3^/µL)	5.32 ± 2.44	4.79 ± 1.71	0.268
Monocytes (×10^3^/µL)	1.047 ± 0.44	1.07 ± 0.39	0.773
Eosinophils (×10^3^/µL)	0.244 ± 0.24	0.23 ± 0.17	0.767
Platelets (×10^3^/µL)	260.60 ± 87.53	258.56 ± 76.52	0.908

* Calculated using an independent sample *t*-test. Mean corpuscular hemoglobin (MCH), Mean corpuscular hemoglobin concentration (MCHC), Mean corpuscular volume (MCV).

**Table 3 biomedicines-11-01651-t003:** Hematological parameters between seropositive and seronegative mothers.

Hematological Parameters	Maternal SARS-CoV-2 IgG Antibodies (*n* = 115)		*p*-Value *
	Seropositive (*n* = 88) (Mean ± SD)	Seronegative (*n* = 27) (Mean ± SD)	
Hemoglobin (g/dL)	15.42 ± 2.06	15.92 ± 1.66	0.252
MCH (pg)	35.37 ± 2.13	35.81 ± 2.19	0.357
MCHC (g/dL)	34.84 ± 1.74	35.13 ± 1.99	0.455
MCV (fL)	101.71 ± 5.22	102.10 ± 5.50	0.741
Neutrophils (×10^3^/µL)	5.97 ± 2.32	6.13 ± 1.92	0.758
Lymphocytes (×10^3^/µL)	2.23 ± 1.01	1.92 ± 0.89	0.156
Monocytes (×10^3^/µL)	3.58 ± 0.37	0.58 ± 0.33	0.996
Eosinophils (×10^3^/µL)	0.11 ± 0.15	0.066 ± 0.11	0.122
Platelets (×10^3^/µL)	251.44 ± 85.74	259.81 ± 68.23	0.644

* Calculated using an independent sample *t*-test. Mean corpuscular hemoglobin (MCH), Mean corpuscular hemoglobin concentration (MCHC), Mean corpuscular volume (MCV).

## Data Availability

Data will be made available upon request.

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
