# Peer review of "Transplacental Transfer of SARS-CoV-2 Receptor-Binding Domain IgG Antibodies from Mothers to Neonates in a Cohort of Pakistani Unvaccinated Mothers"

_biomedicines, 2023, doi:10.3390/biomedicines11061651_

Round 1

Reviewer 1 Report

Overview and general recommendation:

In this study, serum samples from 115 mother neonate pairs were collected and tested for SARS-CoV-2 anti RBD IgG antibodies and hematological parameters. The authors show that there is a strong positive correlation between maternal and neonatal antibodies. The authors also claim that there is no significant statistical difference in the hematological parameters in seropositive and seronegative neonates/mothers.

Overall the manuscript is well written. The approaches and results of the study are clear to readers.  The figures are well organized and presented in an appropriate way. There are still some parts which can be improved. The authors show that SARS-CoV-2 anti RBD IgG antibodies can pass through placenta but they don’t show the biological significances of this process. The result 3.3 and 3.4 are not well analyzed. And I suggest the authors strengthen the importance of this study in the manuscript.

Major comments:

1.      In the introduction part, I suggest the authors include some references which show that some specific IgGs can transfer through the placenta and benefit the neonates to emphasize the importance of this process.

2.      In result part, the authors only show the hematological parameters of seropositive and seronegative neonates/mothers. Did the authors test other physical parameters of neonates/mothers?

3.      In the results part, the authors just display their data but further analysis is not included.

4.      I think the authors should add more discussion about the significance of the research. The main concern of COVID-19 is that it will cause higher percentage of severe illness in old people who is infected.  So how this study will benefit in COVID-19 treatment and prevention? Why this research is outstanding in all the similar researches?

Minor comments:

1.      Page13 ling107, I think it should be “…and completing clinical history…”

Author Response

Thank you very much for reviewing our manuscript. Based on the reviewers’ recommendations, we have thoroughly revised our manuscript. Please see below point-by-point recommendations and our response.

Overview and general recommendation:

Reviewer 1:

In this study, serum samples from 115 mother neonate pairs were collected and tested for SARS-CoV-2 anti RBD IgG antibodies and hematological parameters. The authors show that there is a strong positive correlation between maternal and neonatal antibodies. The authors also claim that there is no significant statistical difference in the hematological parameters in seropositive and seronegative neonates/mothers.

Overall the manuscript is well written. The approaches and results of the study are clear to readers.  The figures are well organized and presented in an appropriate way. There are still some parts which can be improved. The authors show that SARS-CoV-2 anti RBD IgG antibodies can pass through placenta but they don’t show the biological significances of this process. The result 3.3 and 3.4 are not well analyzed. And I suggest the authors strengthen the importance of this study in the manuscript.

Major comments:

  1. In the introduction part, I suggest the authors include some references which show that some specific IgGs can transfer through the placenta and benefit the neonates to emphasize the importance of this process. 

Introduction has been thoroughly revised and contextualized according to the study aims and objectives.

  1. In result part, the authors only show the hematological parameters of seropositive and seronegative neonates/mothers. Did the authors test other physical parameters of neonates/mothers?

We did not analyse other physical and demographic parameters of mothers or the neonates as they were outside the scope of this publication. We did perform a cursory analysis of these parameters but did not find anything interesting. We can include these in the manuscript but that would require significant change in the aims and objectives of the study.

  1. In the results part, the authors just display their data but further analysis is not included. 

We have thoroughly revised our results and the proceeding tables and figures

  1. I think the authors should add more discussion about the significance of the research. The main concern of COVID-19 is that it will cause higher percentage of severe illness in old people who is infected.  So how this study will benefit in COVID-19 treatment and prevention? Why this research is outstanding in all the similar researches?

We have revised parts of the discussion and included more broader discussion on the significance of these findings with respect to disease control and vaccination policies.

Minor comments:

  1. Page13 ling107, I think it should be “…and completing clinical history…”

These have been paraphrased

Please have a look at the revisions. Please do not hesitate to contact me if there are any queries.

Thank you for your patience.

Sincerely,

Dr Yasar Yousafzai (corresponding author)

Associate professor and lab director,

Institute of Pathology and Diagnostic Medicine,

Khyber Medical University,

Peshawar, Pakistan.

Yasar.yousafzai@kmu.edu.pk

Reviewer 2 Report

Thank you for the opportunity to review this work to present if COVID-19 infection at any given time in pregnancy had negative effects on the hematopoietic system of neonates. I appreciated the efforts made by the authors. The work would benefit from some improvements, which I describe below

  • the discussion section and the conclusion section need extensive improvements

Author Response

Thank you very much for reviewing our manuscript. Based on the reviewers’ recommendations, we have thoroughly revised our manuscript. Please see below point-by-point recommendations and our response.

Overview and general recommendation:

Reviewer 2:

Thank you for the opportunity to review this work to present if COVID-19 infection at any given time in pregnancy had negative effects on the hematopoietic system of neonates. I appreciated the efforts made by the authors. The work would benefit from some improvements, which I describe below

  • the discussion section and the conclusion section need extensive improvements

We have thoroughly revised the manuscript including introduction, results and discussion.

Please have a look at the revisions. Please do not hesitate to contact me if there are any queries.

Thank you for your patience.

Sincerely,

Dr Yasar Yousafzai (corresponding author)

Associate professor and lab director,

Institute of Pathology and Diagnostic Medicine,

Khyber Medical University,

Peshawar, Pakistan.

Yasar.yousafzai@kmu.edu.pk

Reviewer 3 Report

Harakeh et al describe a very important observation during the COVID-19 pandemic. It is not only the transfer of maternal antibodies to babies but the health parameters of healthy COVID-19- infected and non-infected non - vaccinated mothers and their babies. Hence, this material is very important.

Method:

“The indirect Enzyme-Linked Immunoassay (ELISA) was performed as per the manufacturer's guidelines using Biotek Elx 800 (USA). 

ELISA was performed using Vircell (REF. No.G1032, LOT.NO.20ECOG114, Vircell, 129 Spain)”

Which one was used or both? 

Table 1:

Is very difficult to understand. My interpretation is that the first part with mother 29.44 years of age delivered pre-term babies with a birth weight of 2.01 kg. This must be clearly described.

5/88 sera did not match between mothers and babies. This is almost 6%. 

“This could be due to technical errors in performing ELISA, gestational age, IgG subclass, and placental pathology” 

It is appreciated that this failure is mentioned. However, the explanation of an ELISA error is very unlikely based on 2 different systems used and placental pathology was excluded. “Gestational age” is not understood by the reviewer.

As stated above, the material collected is very valuable and unique. The reason for this is that COVID-19 is a viral disease that mothers had never encountered in their live. Hence, the immune response is likely a primary but not a secondary immune response like influence, RSV or measles that the mothers have very likely exposed during their life.

Additional information:

Very easy, additional information listed below would be very important as this (remining) material will be lost otherwise over time. Systems for these tests are very well established and largely available:

·      Did the sera of mothers and babies contain COVID-19 RNA detected by PCR? 

·      Did the sera contain neutralizing antibodies? How does the correlation between mothers and babies look like similar to Fig. 2.

·      What are the IgG subclasses (isotypes) that are transferred via the placenta. This might be of particular interest, as the authors have the unique chance to look a primary infection of the mothers as outlined above. 

Author Response

Thank you very much for reviewing our manuscript. Based on the reviewers’ recommendations, we have thoroughly revised our manuscript. Please see below point-by-point recommendations and our response.

Overview and general recommendation:

Reviewer 3:

Harakeh et al describe a very important observation during the COVID-19 pandemic. It is not only the transfer of maternal antibodies to babies but the health parameters of healthy COVID-19- infected and non-infected non - vaccinated mothers and their babies. Hence, this material is very important.

Method:

“The indirect Enzyme-Linked Immunoassay (ELISA) was performed as per the manufacturer's guidelines using Biotek Elx 800 (USA). 

ELISA was performed using Vircell (REF. No.G1032, LOT.NO.20ECOG114, Vircell, 129 Spain)”

Which one was used or both? 

Thanks for your valuable comments and apologies for the confusion. The Biotek ELx 800 is the ELISA reader equipment and Vircell is the ELISA kit. These have been paraphrased and are must clear to read now.

Table 1:

Is very difficult to understand. My interpretation is that the first part with mother 29.44 years of age delivered pre-term babies with a birth weight of 2.01 kg. This must be clearly described.

Apologies for the confusion. Each row represents a separate parameter and columns represent mean and SDs or percentages. We have followed the MDPI template document for making tables and figures.

5/88 sera did not match between mothers and babies. This is almost 6%. 

“This could be due to technical errors in performing ELISA, gestational age, IgG subclass, and placental pathology” 

It is appreciated that this failure is mentioned. However, the explanation of an ELISA error is very unlikely based on 2 different systems used and placental pathology was excluded. “Gestational age” is not understood by the reviewer.

Thank you for your observation and valuable comment. We have revised this part in discussion. In fact, a number of studies have come up with similar observations that show similar proportion of discrepancy between maternal and neonatal seropositivity. We have cited these relevant publications.

As stated above, the material collected is very valuable and unique. The reason for this is that COVID-19 is a viral disease that mothers had never encountered in their live. Hence, the immune response is likely a primary but not a secondary immune response like influence, RSV or measles that the mothers have very likely exposed during their life.

Additional information:

Very easy, additional information listed below would be very important as this (remining) material will be lost otherwise over time. Systems for these tests are very well established and largely available:

  • Did the sera of mothers and babies contain COVID-19 RNA detected by PCR? 

We did not test maternal and neonatal sera for COVID-19 RNA

Did the sera contain neutralizing antibodies? How does the correlation between mothers and babies look like similar to Fig. 2.

Our SARS-COV-2 IgG antibodies kit can detect mix of anti-N and anti-RBD antibodies. There is plenty of evidence to suggest that levels of total IgG antibodies is roughly proportionate to neutralizing capacity of the antibody repertoire. We however, did not test the neutralizing capacity of the sera.

  • What are the IgG subclasses (isotypes) that are transferred via the placenta. This might be of particular interest, as the authors have the unique chance to look a primary infection of the mothers as outlined above. 

Thank you for your comments. We did not test the IgG subclass of the antibodies.

Round 2

Reviewer 1 Report

I think the authors have put considerable effort into addressing the reports of the referees. As a result, the paper is very much improved and I have no problem in recommending it for publication.

Author Response

Thank you for your valuable comments. 

Reviewer 3 Report

If there are sera (mother or child) available COVID-19 they should be tested for the presence of viral specific RNA

Author Response

Thank you very much for your insightful comments. We do not have access to the mother-child serum/ nasopharyngeal swabs for Covid-19 PCR testing. We have acknowledged this as limitation in our discussion.